# Impact of Surface Functionalization by Nanostructured Silver Thin Films on Thermoplastic Central Venous Catheters: Mechanical, Microscopical and Thermal Analyses

**Gregorio Marchiori** [1,†] , **Alessandro Gambardella** [1,†], **Matteo Berni** [2,*] , **Devis Bellucci** [3],
**Giorgio Cassiolas** [4] **and Valeria Cannillo** [3]

1   IRCCS Istituto Ortopedico Rizzoli, Laboratorio di Scienze e Tecnologie Chirurgiche, Via di Barbiano 1/10, 40136 Bologna, Italy; gregorio.marchiori@ior.it (G.M.); alessandro.gambardella@ior.it (A.G.)
2   IRCCS Istituto Ortopedico Rizzoli, Laboratorio di Tecnologia Medica, Via di Barbiano 1/10, 40136 Bologna, Italy
3   Dipartimento di Ingegneria Enzo Ferrari, Università degli Studi di Modena e Reggio Emilia, Via P. Vivarelli 10, 41125 Modena, Italy; devis.bellucci@unimore.it (D.B.); valeria@unimore.it (V.C.)
4   Department of Information Engineering, University of Brescia, 25123 Brescia, Italy; g.cassiolas@unibs.it
*   Correspondence: matteo.berni@ior.it
†   Gregorio Marchiori and Alessandro Gambardella contributed equally to this work as first authors.

**Abstract:** In this work, an interdisciplinary approach was employed to investigate the impact on thermoplastic catheters from the deposition of a thin (180 nm), metallic silver film by a pulsed ablation technique. Our characterization firstly involved tensile and bending tests, each one accompanied by finite element modeling aiming to elucidate the contributions resulting from bulk and coating to the device's mechanical behavior. The morphological assessment of the surface before and after the deposition was performed by atomic force microscopy, specifically implemented to visualize the nanostructured character of the film surface and the extent to which the polymer was modified by the deposition process, focusing on coating delamination due to tensile stress. Finally, thermogravimetric–differential thermal analysis was carried out to evaluate whether silver deposition has affected the physiochemical structure of the polymer matrix. Our results establish that the deposition does not significantly alter the physical and chemical properties of the device. The presented characterization sets a useful precedent for elucidating how structural properties of polymeric materials may change after coating by electronic ablation techniques, highlighting the importance of employing a comprehensive approach for clarifying the effects of additive manufacturing on medical devices.

**Keywords:** central venous catheter; medical device; pulsed electron deposition; nanostructured coating; mechanical testing; finite element modeling; atomic force microscopy; TG-DTA

## 1. Introduction

In the field of biomedical devices, increasing attention is being paid to the usage of nano-fabrication techniques in obtaining specific functionalities related to the biological environment [1,2]. In this perspective, the addition of thin (<1 μm) coatings on specific parts of biomedical devices has been addressed to improve their bioactivity [3] and biodegradability [4] in a cheaper and less time-consuming way than developing completely new biomaterials [5]. In particular, silver (Ag) coatings could be important to give antimicrobial functionality to a specific device without altering its bulk properties [5–11].

The antimicrobial activity of silver has been previously exploited in biomedical devices such as medical and orthopedic implants [7,10,11]; moreover, in recent years, specific studies have been carried out on catheters [8,9] with the aim, at the same time, of avoiding infections and keeping under control silver cytotoxicity when in contact with the biological fluids [9]. In this perspective, particular attention has been devoted to the fabrication of nanostructures—as size reduction allows us to take advantage of novel physical and chemical properties specific to the nanoscale—thus to improve the antibacterial efficacy, and not less importantly, to elucidate the physiochemical mechanism at the base of the antimicrobial action [6,12–14]. Accordingly, the control over the nanoscale features of the surface is often critical to the functional performance of the fabricated layer. Alternatives to the existing deposition techniques call to be explored with the aim of obtaining: (a) the precise control of stoichiometry, morphology and nanostructure; (b) high conformality on complex three-dimensional structures, such as devices or implants, and (c) the possibility of coating heat-sensitive materials [15,16]. To this aim, a wide range of deposition techniques are currently available for addressing specific needs and/or according to available resources [6,13]. Among the physical vapor deposition processes, the potential of pulsed electron ablation (PEA) technique has recently emerged, in particular considering the fabrication of nanostructured thin films addressing biomedical purposes [14–20]. This technique operates at room temperature and is based on the ablation process by means of a highly energetic electron beam of a target surface made of the same material to be deposited. The high non-equilibrium deposition conditions of PEA guarantee preservation of the target stoichiometry and good control upon the physical structure of the films so obtained, making this technique able to meet specific research or application requirements. For example, besides its typical application in depositing ceramics and dielectrics [14–20], this group has demonstrated that PEA was capable of effectively depositing metallic silver ($Ag^0$), preserving the target stoichiometry and allowing to achieve high homogeneity and nanostructuration of the film surface after depositing onto both smooth and rough substrates [21]. Besides prototypical surfaces, the deposition of $Ag^0$ onto three-dimensional devices offers a relevant case of study when applied to the rough and heat-sensitive surfaces of polymers. Due to their low cohesion energy, these materials exhibit lower surface energy with respect to the metals [22] and, as a consequence, tend to promote the self-aggregation of the species deposited atop; in particular, metallic films deposited on polymers tend to form clusters and/or islands, thus preventing nanostructuration [23]. Moreover, the main factors influencing the thin film deposition—i.e., temperature, pressure, and energy of the deposited species—also affect the integrity of the polymer [24]. Thus, possible effects of the deposition process on the treated materials should be verified [25,26]. This study concerns the functionalization of commercial polymeric central venous catheters (CVCs) by metallic Ag coating achieved through PEA. In particular, our focus is to elucidate the impact of the functionalization on the morphological characteristics of the surface and on the mechanical behavior of the device. Mechanical tests presented here involve tensile and bending experiments, in accordance with the intravascular catheters standard [27] and usage, which are compared to the relative results achieved by finite element modeling (FEM) simulations. Atomic force microscopy (AFM) is used to investigate the surface before and after deposition, thus to verify the extent to which the nanostructuration of the surface is obtained; moreover, since tensile stress caused partial coating delamination, AFM offers the opportunity to investigate the effect of the deposition on the polymer surface and to compare this result with the morphological characteristics of the device before the deposition. Finally, thermogravimetric–differential thermal analysis (TG–DTA) was carried out to elucidate the extent to which any difference in the mechanical response could be attributed to changes in the superficial structure or in the thermal behavior of the polymer.

## 2. Materials and Methods

### 2.1. Central Venous Catheter

Novel double-lumen Central Venous Catheters (CVCs, B.Braun Avitum Italy S.r.l, Mirandola, Italy)—obtained through the processing of thermoplastic polyurethanes (PU) named Carbothane

(Lubrizol-Lubrizol LifeScience, Cleveland, OH, USA)—represent the here investigated medical device (Figure 1, side view). CVC main dimensions are: outer diameter $\emptyset_o = (4.30 \pm 0.05)$ mm, inner diameter $\emptyset_i = (3.40 \pm 0.05)$ mm, septum $s = (0.20 \pm 0.05$ mm).

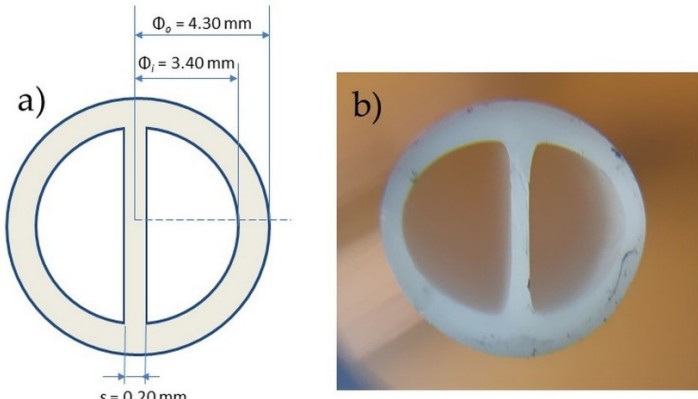

**Figure 1.** (**a**) Side view of the CVC with indication of the outer ($\emptyset_o$), inner ($\emptyset_i$) and septum ($s$) dimensions. (**b**) Magnificated picture of the CVC section.

## 2.2. Silver Deposition

Ag thin films were deposited by PEA based on a commercially available system (Noivion Srl, Rovereto, Italy). In such a technique, a high voltage pulsed electron beam (amplitude of up to 25 kV and short period, <1 μs) causes the emission of species from a target surface made of the same material to be deposited (ablation process), with consequent formation of a plasma plume; this latter is addressed towards the substrate surface, where a thin material film is formed atop. Compared to traditional PEA systems, the geometrical configuration used here allows to obtain an increased impact of the electron beam onto the target surface, leading to an increased efficiency of the ablation process and, consequently, a higher deposition rate [28,29]. In the present study, ablation occurred at the surface of a rotating cylindrical 99.99% pure silver target ($\emptyset = 25.4$ mm, thickness = 3.18 mm, Kurt J. Lesker Company Ltd., Jefferson Hills, PA, USA) by using an electron beam pulse of duration $\tau = 100$ ns, energy $E \approx V^2 \approx 10$ J and power density $P \approx 10^9$ W/cm$^2$. The vacuum chamber was initially evacuated down to a pressure of $1.0 \times 10^{-5}$ mbar by a turbo-molecular pump (EXT255H, Edwards, Crawley, UK) and then raised by a controlled flow of oxygen (purity level = 99.99%) to $2.0 \times 10^{-4}$ mbar. Based on our previous work, the deposition could be addressed to obtaining good quality films with nanostructured surface; moreover, X-ray Spectroscopy Photoemission (XPS) analyses including the measurement of the Auger parameter allowed to establish that only metallic silver ($Ag^0$) was deposited [30]. Depositions were carried out simultaneously on CVCs and silica slices (size $10 \times 10 \times 1$ mm$^3$, Fondazione Bruno Kessler, Trento, Italy). As CVCs were received in sealed package from the manufacturer—i.e., B. Braun Avitum Italy S.r.l—after sterilization in accordance to the relative normative of such medical device, no further preparation/cleaning of the surface was required before their insertion in the vacuum chamber. The film thickness $h = (180 \pm 10)$ nm was measured by AFM after depositing on silica slices. To guarantee a uniform and homogeneous film deposition, both CVCs and silica slices were placed on a custom-made mobilization system specifically designed for multiple samples depositions (Figure 2). While a principal coupling of gears transfers the rotation from the engine to the samples holder—so that each sample periodically comes closer to the focus of the plasma plume (Figure 2a)—secondary gears allow self-rotation of each sample, so the whole external surface of the device is coated. All the components of the mobilization system were fabricated in Polylactic Acid by 3D printing.

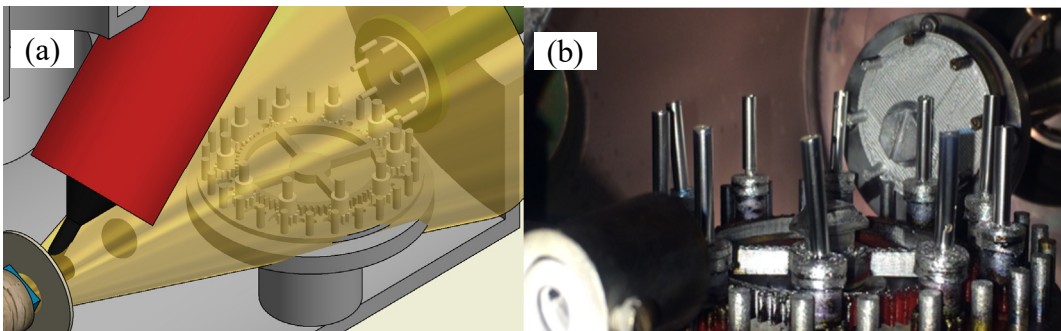

**Figure 2.** Mobilization system for the deposition by PEA onto CVCs. (**a**) CAD axonometric view of the deposition camera, with the plasma plume (yellow) that propagates from the target surface toward the CVCs; these latter are placed in vertical position at regularly spaced holders on the internal rotating gear, thus to cover only their external surface (**b**).

### 2.3. Uniaxial Tensile Test

#### 2.3.1. Experimental

Uniaxial tensile tests were performed on a material testing machine (Mod. C42, MTS, Eden Prairie, MN, USA), according to CVCs' related Standard [27]. The tested devices—six for both uncoated (control) and coated group—were clamped at their opposite ends by a pair of drill chucks [31] to achieve a clamp-to-clamp distance of $L = (80 \pm 2)$ mm. All tests were then carried out at room temperature with an extension rate of 220 mm/min, until failure occurred [27,31]. The force-displacement sampling rate was 2 Hz. Even though the physiological failure of the investigated devices would rather be caused by fatigue (i.e., cyclic stress) and/or bending, monotonic tests provide a reliable and simple comparison to detect changes in their mechanical behavior [31]. Linear stiffness ($S_T$, subscript $T$ indicates tensile loading), ultimate force ($F_u$) and displacement ($d_u$) were estimated for both uncoated and coated samples groups.

#### 2.3.2. Finite Element Modeling

FEM (Abaqus, Simulia, Johnston, RI, USA) was used to investigate the initial $S_T$ of the CVC under tensile loading, aiming to highlight the different contributions of substrate and coating. Simulations were carried out considering the device dimensions specified in the previous sections—i.e., $\varnothing_o$, $\varnothing_i$, $L$ and $s$ for the CVC, h for the coating. To simulate uniaxial tensile test, the lower boundary of the device was grounded, while the upper part was constrained in a 10 mm vertical displacement; the vertical force was then evaluated as output in a static-mode simulation. The coating was modelled as perfectly adhered to the substrate. Mesh elements were 8-node bricks with average size of 0.2 mm for the bulk and 8-node thick shells with average size of 0.2 mm for the coating. Both the bulk and coating materials—i.e., Carbothane and Ag—were modelled as isotropic linear elastic materials, with Poisson's ratio fixed at 0.30. Within this modelling, beyond to Poisson's ratio, the material is defined by the Young's modulus ($E$), which is related to the stiffness in the case of a solid body. Aiming to determine separate values of E for the considered materials—i.e., Carbothane ($E_C$) and Ag ($E_{Ag}$)—, the Young's modulus of both was varied until FEM simulation reached the experimental value of $S_T$, first considering the samples before deposition and, then, after this process. At this point, FEM was used to: (a) investigate the sensitivity of $S_T$ respect to the coating properties and (b) establish the bases for simulating the bending test.

*2.4. Bending Test*

2.4.1. Experimental

Bending tests were performed on a multiaxial mechanical tester (Mach-1 v500css, Biomomentum, Laval, QC, Canada) aiming to quantitative reproduce ASTM B571 Standard, which suggests to evaluate the adhesion of metallic coating after bending the device [32]. Specifically, bending was implemented in a "three-point" configuration, where supporting and loading aluminium pins had a diameter of four times the thickness of the CVC, i.e., 17.2 mm. A sketch of the testing configuration is reported in Figure 3. When the loading pin reached contact with the device, it moved until the two specimen's "legs" resulted parallel; this configuration was reached by imposing a vertical translation of 26 mm.

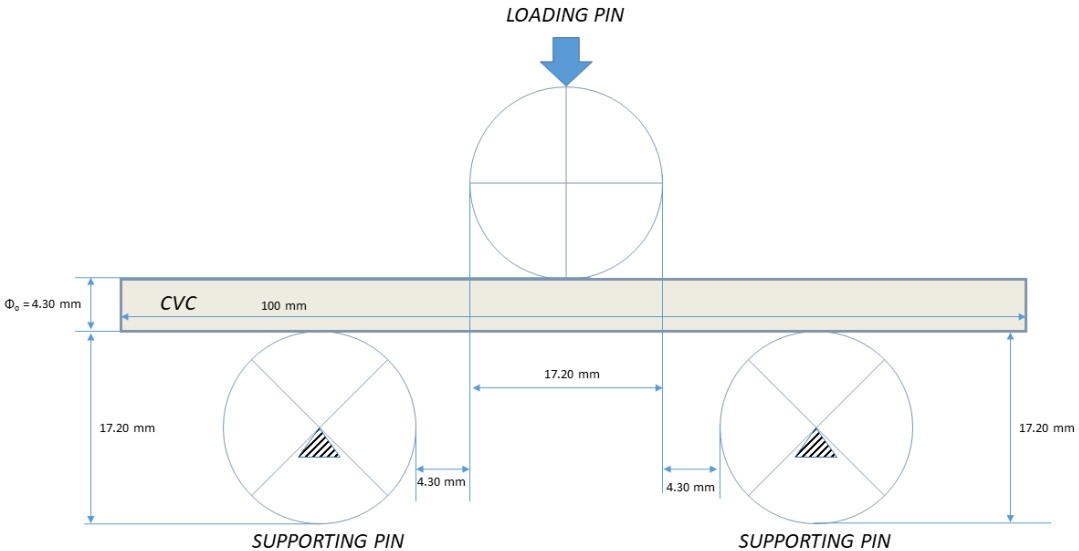

**Figure 3.** Side view of the three-point bending configuration: supporting pins are fixed, loading pin moves down on the plane.

Six samples for both uncoated (control) and coated group were tested. Tests were carried out at room temperature with a motion rate of the loading pin of 220 mm/min. The force-displacement sampling rate was 2 Hz. Linear stiffness ($S_B$, subscript B indicates bending) was determined.

2.4.2. Finite Element Modeling

FEM was used to separate the different contributions of coating and friction in affecting SB. Indeed, material parameters were fixed to the values validated through tensile test simulation, while coefficient of friction (CoF) between CVC and pins was free to vary. Mesh parameters were the same as of tensile simulation. Finally, FEM was used to compare coating stress between tensioning and bending at the same level of displacement, i.e., 10 mm corresponding to the limit of the tensile simulation.

*2.5. Atomic Force Microscopy*

CVCs were imaged in tapping mode by using a stand-alone microscope (NT-MDT Co., Moscow, Russia) operating in air at room temperature, equipped with Si cantilevers (tip curvature radius ≈ 10 nm and resonant frequency ≈ 240 kHz). The topographies were recorded at resolution of 512 × 512 points on several non-overlapped regions. All images were reported unfiltered, except for a 2nd order leveling.

*2.6. Thermogravimetric and Differential Thermal Analysis*

Thermogravimetric and Differential Thermal Analysis (TG-DTA) were performed to assess the thermal behaviour of uncoated and coated CVCs. A Netzsch Differential Thermal Analyzer STA

429 CD (NETZSCH-Gerätebau GmbH, Selb, Germany) was used for testing a 15 mg sample for each typology—i.e., uncoated and coated—heated from room temperature to about 800 °C (heating rate 10 °C/min). The tests were performed in a transient environment in air at a flow rate of 50 mL/min.

*2.7. Notations and Statistical Analysis*

$S_T$, $F_u$, $d_u$ and $S_B$ were divided into uncoated and coated groups and displayed as mean value ± standard deviation. Statistical difference between groups was then assessed by using the two-sided Wilcoxon rank sum test with a level of statistical significance $\alpha = 0.05$ ($p = 0.05$). Considering the analysis of the results achieved by FEM (FEA)—and aiming to obtain an indication about threshold of coating $h$ and $E_{Ag}$ ($h^*$ and $E_{Ag}^*$, respectively) above which difference in $S_T$ between control and coated groups becomes statistically significant—the following expression of the $t$ parameter in the t-test is used:

$$t = \frac{X_1 - X_2}{\sqrt{\left(\frac{(N_1-1)\sigma_1^2 + (N_2-1)\sigma_2^2}{N_1+N_2-2}\right)\left(\frac{1}{N_1} + \frac{1}{N_2}\right)}} \tag{1}$$

where $X_1$ and $X_2$ are related to coated and uncoated samples, $\sigma_1$ and $\sigma_2$ are the corresponding standard deviations, $N_1$ and $N_2$ the samples size of each group.

## 3. Results and Discussion

*3.1. Effect of the Mechanical Stress on the Coating*

After the thin film deposition, the device surface appears uniformly covered (Figure 4b). In the zoomed optical image of Figure 4b is evident how the fabricated film conformally follows the macroscopic extrusion lines of the underneath polymer. Moreover, the right side of Figure 4 also resembles the effect of the mechanical tests on the coating; tensioning (Figure 4c) causes important coating delamination, revealing the polymer surface underneath, while bending (Figure 4d)—which is by far the most common solicitation occurring during normal catheter operation—appears not to affect the integrity of the coated surface. In the next paragraphs we focus on the numerical aspects of the mechanical tests and simulations, aiming to separate contributions of Carbothane and Ag on the device mechanical behavior; henceforth, morphological aspects of the device, considered before and after the deposition, will be examined in the paragraph dedicated to AFM investigation.

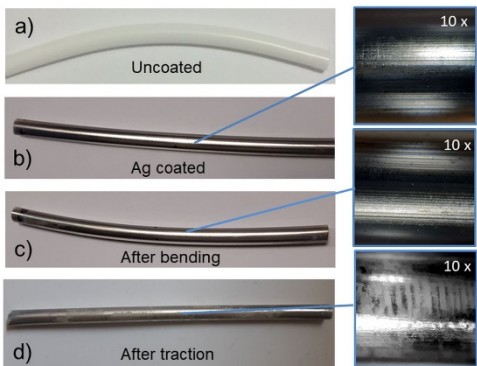

**Figure 4.** Optical images of the CVC before (**a**) and after (**b**) deposition. Moreover, optical images of the coated device were also acquired after tensile (**c**) and bending test (**d**). On the right, magnified images highlight details of the respective surfaces.

*3.2. Uniaxial Tensile Tests and FEA*

Experimental values of $S_T$, $F_u$ and $d_u$ for each group (uncoated and coated), estimated according to Section 2.3.1, are reported in Table 1. Coated samples exhibited slightly lower % dispersion of the data compared to uncoated ones. Despite of the absence of statistically relevant difference in $S_T$

($p$ = 0.6991), $F_u$ ($p$ = 1) and $d_u$ ($p$ = 0.3939) between uncoated and coated samples, a slight increase in rigidity and strength of the coated samples was observed.

**Table 1.** Tensile stiffness ($S_T$), ultimate force ($F_u$) and displacement ($d_u$) for each evaluated group (uncoated and coated). The percentage represents the coefficient of variation (CoV%) of the estimated parameters.

| Group | $S_T$ (N/mm) | $F_u$ (N) | $d_u$ (mm) |
|---|---|---|---|
| Uncoated | 9.47 ± 1.73 (18.3%) | 165.1 ± 8.6 (5.2%) | 184.9 ± 6.6 (3.6%) |
| Ag Coated | 9.92 ± 1.15 (11.6%) | 165 ± 4 (2.5%) | 189.2 ± 6.9 (3.7%) |

The $S_T$ values reported in Table 1 were then used as target of FEM simulations to obtain the corresponding $E_C$ and $E_{Ag}$ values (see Section 2.3.2). In this respect, $E_C$ = 143.4 MPa and $E_{Ag}$ = 13.0 GPa were obtained in correspondence of 9.47 N/mm (uncoated) and 9.92 N/mm (coated samples), respectively. $E_C$ and $E_{Ag}$ resulted respectively higher and lower of a factor with respect to the values reported in literature for the corresponding materials, i.e., Carbothane [33] and Ag [34]; nevertheless, they still differ each other of an acceptable amount (~90 times). As regard to $E_C$, large variability is suggested by the literature among the different compositions of Polyurethanes [35]. Concerning $E_{Ag}$, it is interesting to investigate the reasons behind the discrepancy between the here estimated value of the coating and the Young's modulus reported in literature for the bulk metal. To this aim, it is opportune to estimate the extent to which the difference between the stiffnesses of the uncoated and coated CVC become statistically relevant. For example, by assuming $p$ = 0.05—which corresponds to a $t$ = 2.571 according to the tabular value of the two-tails test—it would result $S_T^*$ = 11.65 N/mm for the coated group. If FEM analysis is then repeated, $S_T^*$ ($E_{Ag}$ = 13 GPa) corresponds to $h^* \sim 5 \cdot h$ = 900 nm, while $S_T^*$ ($h$ = 180 nm) corresponds to $E_{Ag}^* \sim 5.6 \cdot E_{Ag}$ = 73 GPa, which agrees very well with literature [34]. Therefore, $E_{Ag}$ is as such that an ~1 µm-thick coating is needed to significantly increase $S_T$ respect to the uncoated device; otherwise, considering unvaried the coating thickness ($h$), an increased stiffness would require that the Young's modulus of Ag-coating is higher, and precisely close to the bulk limit [34]. Possible reasons behind these findings could concern approximations related to the FEM approach; nevertheless, micro-structural aspects related to the film formation appear more likely to be considered and, thus, will be detailed in the following paragraphs. If a 10% variation of the thickness is considered, FEM can provide the corresponding variation of $S_T$, as reported in Table 2.

**Table 2.** Effect of the coating thickness ($h$) variation on the CVC stiffness ($S_T$), assessed through FEM.

| - | Case | $S_T$ (N/mm) |
|---|---|---|
| | −10% $h$ | 9.87 |
| $h$ = 180 nm | $h$ | 9.92 |
| | +10% $h$ | 9.96 |

A linear relation ($R^2$ = 0.996) between the values of Table 2 can be inferred (note that, by definition, it results $S_T$ ($h$ = 0) = 9.47 N/mm) and, thus, taken as a good approximation of $S_T(h)$ for the geometry of the coated device where, in general, the presence of two different Young's moduli could not be easy to account theoretically.

### 3.3. Bending Tests and FEA

Experimentally determined $S_B$ resulted (0.36 ± 0.02) N/mm for the uncoated group and (0.40 ± 0.04) N/mm for the coated one; no statistically significant difference was found between them ($p$ = 0.2286). Moving to the relative FEM, the experimental $S_B$ of the uncoated device was obtained by assuming a polymer-aluminium CoF of 2.0. If the latter value of CoF was assumed in case of the coated

device, the corresponding $S_B$ resulted 0.38 N/mm, hence lower by a slight amount respect to the experimental one. If experimental values for the coated CVCs would be lowered by that amount, the difference between uncoated and coated groups would be still lower ($p = 0.8571$), highlighting that a contribute to the stiffening could be attributed to an increase in friction. In fact, considering coated CVC, FEM simulation reached the experimental $S_B$ by imposing a silver-aluminium CoF of 2.5%, 25% higher than the polymer-aluminium CoF condition.

Concerning the stress at which the coating was subjected during the tests, and considering a 10 mm vertical displacement, simulated bending highlighted a load of 3.3 N and a maximum principal stress of 1114 MPa acting on the coating, specifically confined on the mid-region of the device, while simulated tensile test showed a load of ~100 N and a maximum principal stress of 1756 MPa affecting a broader area of the device surface; these evidences partially explain why the film detachment resulted more evident after tensioning rather than bending.

### 3.4. Atomic Force Microscopy

AFM proved to be a powerful tool for investigating polymer surface at sub micrometric scale and, thus, for comparing its initial and final morphology under mechanical stress [36–40]. In our case, such technique represents the ideal tool for detailing modifications of the surface occurring sub-micrometer scales and for retrieving quantitative parameters. Representative images of the CVC before and after deposition are shown in Figure 5. The uncoated device (Figure 5a) exhibited morphological elements consisting of spherulites; secondarily, rod-like segments randomly distributed outside the spherulites are here indicated by blue lines.

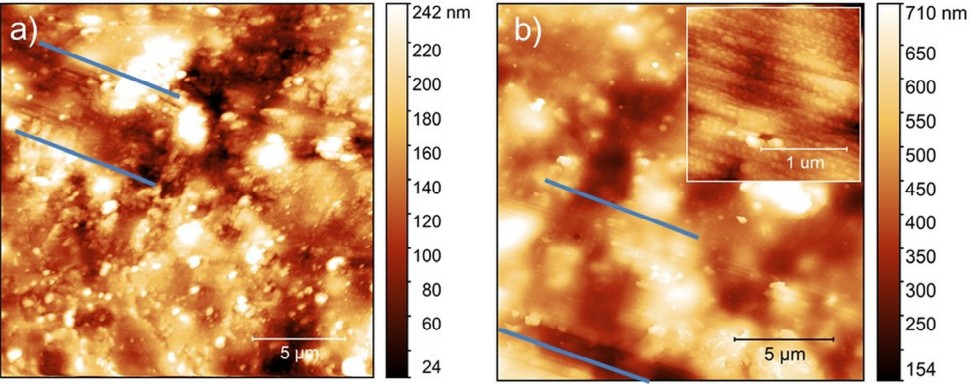

**Figure 5.** $20 \times 20 \ \mu m^2$ AFM images of the surface before (**a**) and after (**b**) Ag deposition. Blue lines in Figure 5a indicate a fibre-like region; the latter regions resulted conformally coated by the silver nanostructure, as shown in Figure 5b and in the zoomed-in $2 \times 2 \ \mu m^2$ image (inset).

After deposition, the presence of the homogeneous metal layer covering the polymer surface is revealed by the increased peak-to-peak values (Figure 5b); this is confirmed by the bolded values in Table 3, which evidence how the surface roughness increases after the deposition, likely due to the aggregation of metal clusters in correspondence of the more complex topological features of the polymer, which act as accumulation sites for particles impinging the substrate [41]. This circumstance is favored by the expected low mobility of the deposited species, which takes place during the process due to an inherent shadowing effect [19,21]. Further magnification of the coated surface enables the observation of a densely packed texture of nanoclusters, locally ordered along the direction of the segments (Figure 5b).

**Table 3.** Root mean square roughness, *R*s (nm), corresponding to uncoated and Ag-coated samples before and after tensile test. All the reported values were extracted from $20 \times 20\ \mu m^2$ images.

| Sample | Before Tensioning | After Tensioning |
|---|---|---|
| Uncoated | 54.6 ± 8.2 | 56.3 ± 6.6 |
| Coated | 94.6 ± 14.7 | 100.5 ± 30.2 |
| Delaminated | - | 75.8 ± 5.6 |

These findings are in good agreement with our previous results obtained during silver growth by PEA onto non-smooth and non-prototypical substrates [30]. Moreover, in the present case, the strong inhomogeneity of the substrate morphology may lead to a non-uniform distribution of the clusters during the growth of the film, resulting locally in strong variations of the film thickness; in other words, the rough substrate influences the microstructural process of formation of the film during growth, resulting in a Young's modulus lower than expected [34]. Due to the most evident solicitation, and thus delamination, produced by tensioning, it is interesting to investigate whether the coating and the underlying polymer surface have been altered in their morphology as a consequence of such a stress and of the deposition process. Representative AFM images of the device after tensioning, considering both coated and delaminated regions, are shown in Figure 6. Moreover, the corresponding $R_S$ values of such areas are reported in the last column of Table 3.

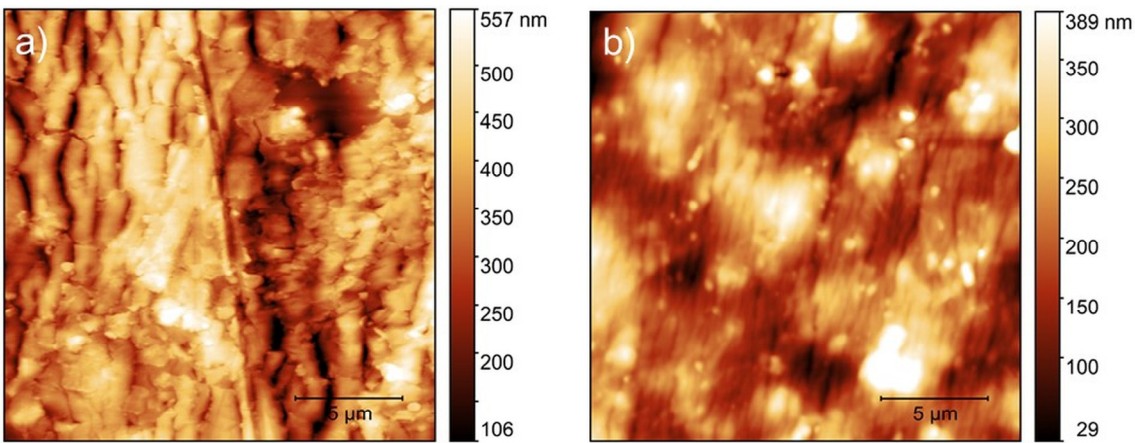

**Figure 6.** AFM images of the coated (**a**) and delaminated (**b**) CVC surface after tensioning.

Ripples oriented perpendicular to the elongation direction have formed across the coated surface (Figure 6a); the wavy pattern so obtained is accompanied, not surprisingly, by a roughness increase with respect to untested samples (Table 3). Coating delamination is also observed at microscopic scale, as evidenced in Figure 6b; remarkably, also the delaminated surface exhibits vertical stripes as a result of the polymer surface relaxation after tensioning; the observed morphology can be justified well in terms of buckling instability of the coating, which depends on both the film thickness and the rigidity of the polymer [37]. By looking at Table 3 (underlined values), the influence of the coating process on the polymer surface is evidenced by the increased $R_S$ of the delaminated regions respect to the uncoated device, both considered after tensioning. Thus, the metal coating influenced the soft polymer surface and, by applying external mechanical stress, was able to cause important superficial morphological modifications.

*3.5. Thermogravimetric and Differential Thermal Analysis*

TG-DTA curve of uncoated CVC is shown in Figure 7a. There is an endothermic peak at 360 °C. A small exothermic peak at 273 °C can be observed. A more pronounced exothermic peak at 440 °C can be observed as well, with two shoulders, one at 380 °C and the other at 478 °C. This latter exothermic

peak coincides with a significant mass loss signal. At 538 °C another exothermic peak corresponds to the final decrease of sample mass.

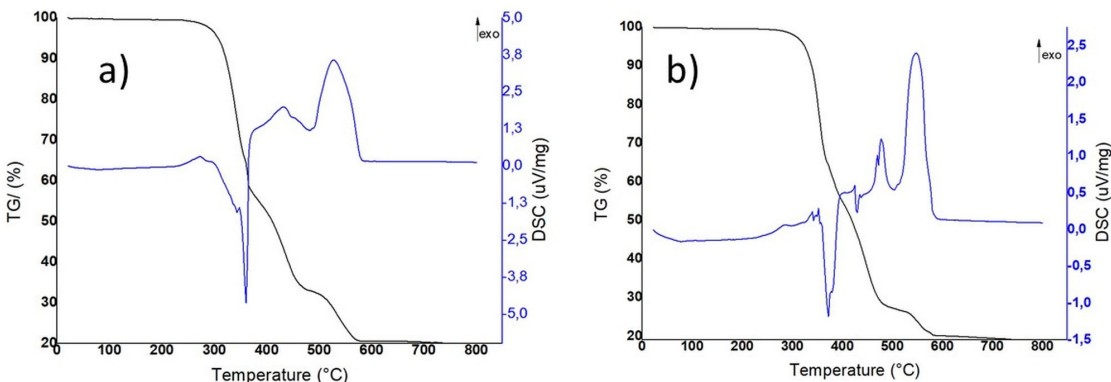

**Figure 7.** TG-DTA curve of the uncoated CVC (**a**) and coated CVC (**b**).

The TG-DTA curve of the coated CVC is shown in Figure 7b, in which it is possible to observe an endothermic peak at 372 °C. A small exothermic peak at 269 °C and a much more pronounced exothermic peak at 427 °C can be also detected; such peaks are equivalent to those reported in Figure 7a for the uncoated CVC. However, at about 480 °C there is a pronounced exothermic peak, which is absent in the uncoated CVC. At 552 °C another exothermic peak corresponds to the final decrease of sample mass. Regarding the polymeric matrix considered as a whole, the TG-DTA curve of coated samples (reported in Figure 7b) showed a similar trend to that of uncoated device, even if some differences are detectable. While the TG curves are almost identical, the DTA curve points out some discrepancies. Such a difference could be probably due to the silver coating, which slightly influences the thermoplastic PU behavior, as reported for example in literature for another polymer [42]. Anyway, the differences are not so relevant, and it is possible to conclude that both the deposition process and the coating itself do not significantly affect the device thermal behavior. Different decomposition curves of thermoplastic polyurethanes can be found in the literature. Reports can be different not only because of differing thermoplastic polyurethanes composition but also due to the variety of different testing conditions, such as heating rates and atmosphere (e.g., nitrogen [43], or air [44]). Nevertheless, no marked difference resulted when thermogravimetric analyses were run in nitrogen or air [44]. Concerning uncoated samples, the major exothermic peak is shifted compared to major exothermic peak reported in literature [44,45]. Such a shift can be ascribed to different thermoplastic polyurethane compositions; however, both the trend of DTA curve and samples mass loss showed here are similar to the curves reported in literature [46]. Additionally, a complete burnout of the thermoplastic polyurethane residue takes place between 500 °C and 595 °C, as showed by the exothermic peak. This is in line with thermal behaviour reported in literature [47], in which the complete burnout takes place between 500 °C and 600 °C in air.

### 3.6. Comparison with Similar Studies

In the same field of this work, previous studies investigated the functionalization of a polymeric catheter surface by antibacterial silver coating [48–53]. Nevertheless, authors generally focused on the coating—assessed through in vitro analysis or in vivo trials—but no specific information about substrate modification was reported, with the exception of Pollini et al. [52] and Wu et al. [53], which, however, provided respectively EDX and XPS spectra of both control and coated devices with no further justification. The influence of the functionalization process on the substrate was instead partially discussed in the deposition of metallic coatings on polymeric surfaces [54,55]. In particular, Lupoi et al. [54] investigated the potential of the cold spray process in producing metallic coatings onto polymers and composites. Despite of the process optimization, a heavy erosion of the substrate was found as a result of the particles impact on the polymeric surface. Chena et al. [55] explored the possibility

of metalizing PEEK surface through cold spray, suggesting that the high temperature impinging gas makes rapid softening of the polymer substrate upon heat exchange. The high impact speed and temperature of metallic powder can definitely produce serious erosion of the polymer substrate. Both these last studies partially investigated substrate modification through analysis at the microscopic level. Accordingly, future works should be directed in deepening the combination of analyses at the micro- and macro-scale, thus to correlate alterations at the microscopic level to relative changes in the macroscopic behavior of the device. Moreover, this study focused on chemical-physical-mechanical properties of the coated medical device. Further analyses will investigate aspects more closely related to the coating functionality, such as biocompatibility and cytotoxicity.

## 4. Conclusions

In this study an interdisciplinary approach was used to assess modifications of a polymeric central venous catheter (bulk and surface) after the deposition of nanostructured metallic Ag thin films. Small but not significant differences were found in the experimental mechanical parameters—i.e., stiffness, ultimate force and ultimate displacement in tensioning, and stiffness in bending—between uncoated and coated devices; in this respect, finite elements analysis suggested that the slight changes observed in the device response could be entirely ascribed to the presence of the coating. Moreover, it is reasonable to suggest that the inhomogeneous aspect of the coated surface revealed by atomic force microscopy, originated in turn by the strong inheritance of the substrate morphology, could be the reason at the basis of the observed lowering of the coating's Young's modulus with respect to bulk Ag. Thermogravimetric Differential Thermal Analysis confirmed that no alteration occurred in the polymeric matrix thermal stability. The present study suggested that, in developing functionalized surfaces for medical purposes, a comprehensive investigation of the final devices must be taken into account, not just to evaluate the potential benefits conferred by the treatment, but also to assess the effect of the process on the medical devices themselves.

**Author Contributions:** Conceptualization, G.M. and M.B.; Data curation, G.M., A.G., M.B., and V.C.; Investigation, A.G., M.B., D.B., G.C., and V.C.; Methodology, G.M., A.G., M.B., D.B., G.C., and V.C.; Validation, G.M., A.G., M.B., and V.C.; Writing—original draft, G.M., A.G., M.B., and V.C.; Writing—review & editing, G.M., A.G., and M.B. All authors have read and agreed to the published version of the manuscript.

**Funding:** This research was funded by POR-FESR 2014–2020 project: "NANOCOATINGS–Nuovi film antibatterici nanostrutturati per applicazioni in campo biomedicale".

**Acknowledgments:** The authors are grateful to R&D Department of B. Braun Avitum Italy (Mirandola) for scientific and experimental support. AFM measurements were carried out at the facilities of ISMN-CNR@spmlab.

**Conflicts of Interest:** The authors declare no conflict of interest.

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
