# Peer review of "Impact of Surface Functionalization by Nanostructured Silver Thin Films on Thermoplastic Central Venous Catheters: Mechanical, Microscopical and Thermal Analyses"

_coatings, doi:10.3390/coatings10111034_

Round 1

Reviewer 1 Report

The manuscript is well written and presents interesting results, but some minor text editing is required (for example lines 131, 276). I suggest to accept in the present form after minor text correction.

Author Response

The manuscript is well written and presents interesting results, but some minor text editing is required (for example lines 131, 276). I suggest to accept in the present form after minor text correction.

We thank the Reviewer for the positive comment. We have performed text corrections as required.

Reviewer 2 Report

The manuscript describes the impact of surface functionalization by nanostructured silver thin films on thermoplastic central venous catheters: mechanical, microscopical and thermal analyses.

The covering of thermoplastic central venous catheters with silver could be important because this metal have special properties. However, the reactivity of silver is important and it is not taken into account. What is happen with silver in biological fluids? Why the implants are not usually made from silver?

The authors must explain better the choice of silver as coverage layer. Why not titanium, zirconia or other well-known metal used in implantology was not used?

In Introduction are not presented previous studies in the field of implantology in medicine. Please improve.

The paper is not really new because the deposition of Ag by PEA was already carried out. Please explain.

The characterization of the layer by AFM is a relative small importance. Why SEM was not employed?

Thermogravimetric-Differential Thermal Analysis was employed for study the changes on the mechanical response. Why not typical tests for mechanical properties were not employed.

The study of interactions with biological fluids should be included in the manuscript.

Lines 207-208. Several subscript characters are necessary.

In Figure 1 an image of the CVC should be included.

Figure 3 is not relevant and it can be removed from the manuscript.

All the ideas about the use in medicine are not supported by the information provided in the manuscript.

Author Response

The manuscript describes the impact of surface functionalization by nanostructured silver thin films on thermoplastic central venous catheters: mechanical, microscopical and thermal analyses.

The covering of thermoplastic central venous catheters with silver could be important because this metal have special properties. However, the reactivity of silver is important and it is not taken into account. What is happen with silver in biological fluids? Why the implants are not usually made from silver? The authors must explain better the choice of silver as coverage layer. Why not titanium, zirconia or other well-known metal used in implantology was not used?

In Introduction are not presented previous studies in the field of implantology in medicine. Please improve.

The paper is not really new because the deposition of Ag by PEA was already carried out. Please explain.

The study of interactions with biological fluids should be included in the manuscript.

All the ideas about the use in medicine are not supported by the information provided in the manuscript.

We thank the Reviewer for giving us the opportunity to better discuss these fundamental aspects.

The rationale behind the use of a nanometric silver coating is to give antimicrobial properties to the device without altering this latter’ bulk properties; the antimicrobial activity of silver has been previously exploited in both orthopedic implants (Ref. 7 now added as new in the main text) and biomedical devices – about the latter specifically on catheters (Ref. 8 now added as new in the main text) - with the aim, at the same time, of avoiding infections and keeping under control silver cytotoxicity when in contact with the biological fluids (Ref. 9 now added as new in the main text). In this context, in our work we focused mainly on the opportunity offered by the deposition technique, which is used for the first time with this purpose, to obtain nanostructuration of the surface, as nanostructuration is crucial for the efficiency of the device as antimicrobic (this, of course, implies that further attention must be payed to control the consequent cytotoxicity of the coating (Ref. 12 now added as new in the main text). In our previous work (Ref. 30 in the main text) we deposited silver coatings by PEA on prototypical substrates materials, and focused on how obtaining nanostructuration and high purity of the metal surface; nevertheless, the work presented here represents a second step ahead, i.e. the application of the deposition technique to a real medical device. We are aware that the interaction with biological fluids must be object of careful evaluation (as stated in the main text, Lines 50 to 53 of the Introduction); however, this is beyond the scope of the present work which represents an intermediate (and, in our opinion, necessary) step of knowledge. Indeed, preliminarily to the phase of the biological assays, it is mandatory to verify that the deposition process did not alter significantly the base properties of the original device, and to understand whether the films nanostructuration was preserved after deposition onto a substrate with a complex morphology and high aspect ratio such as the CVC surface.

We have now added the references of above and modified some sentences in the Introduction (Lines 46 to 53) to improve the background of the manuscript and better address our purposes and the Reviewer’s concerns; the overall content of the manuscript has also been modified in this sense, hoping to highlight the quality of our work.

The characterization of the layer by AFM is a relative small importance. Why SEM was not employed?

As the recognition of the silver nanostructure is important for our purposes, topographic measurements at high spatial resolution were required. In this respect, as the size of the single grains does not exceed a few tens of nm’s, AFM is the ideal tool to use. A further reason was the possibility of testing roughness variations before and after mechanical tests, as reported in the manuscript. All these quantitative information could not be retrieved from SEM images; we have now better evidenced this circumstance in the manuscript (Lines 349-351 of the section 3.3. Atomic Force Microscopy).

Thermogravimetric-Differential Thermal Analysis was employed for study the changes on the mechanical response. Why not typical tests for mechanical properties were not employed.

We thank the Reviewer for the observation. Thermogravimetric-Differential Thermal Analysis was here adopted to investigate any structural changes eventually produced inside the polymer, thus, to explain alterations that might have occurred in the device mechanical response as a consequence of its exposure to the deposition process. The information provided by such a technique was consistent with the macroscopic response investigated by the here realized mechanical tests, i.e. tension and bending test, which have the aim of reproducing the stress at which the medical device is subjected to during its in vivo usage.

Other typical mechanical tests could have been employed to investigate the specific mechanical properties of both polymer and coating at the nano and microscopic scale, first and foremost indentation. Nevertheless, such tests would have required specifications of the sample, e.g. planarity, very difficult to achieve in the case of a geometrical complex device such as a catheter. Aiming to overcome these issues, Finite Element Simulations were here conducted to reveal specific material properties, thus to highlight the influence of the coating on the mechanical response of the device and the stress at which the coating is subjected.

Lines 207-208. Several subscript characters are necessary.

We apologize for these mistakes; all these items have been fixed.

In Figure 1 an image of the CVC should be included.

We have now updated Figure 1 accordingly.

Figure 3 is not relevant and it can be removed from the manuscript.

Although the three-point bending technique is well known, we preferred to maintain Fig. 3 as it reports some relevant dimensions of the system which can be helpful to the experimentalist for repeating the tests.

Reviewer 3 Report

The manuscript ‘Impact of surface functionalization by nanostructured silver thin films on thermoplastic central venous catheters: mechanical, microscopical and thermal analyses’ is interesting and written well. It can be accepted after a minor correction based on the below comments.

1.Line 51: of obtaining…

2.Line 64: Ag0…0 can be in superscript, or else it looks like silver oxide.

3.Sec. 2.2.: The coating is done on the top only or including in the septum region also?.

4.Line 109: The oxygen was used in the chamber thus the coating may be in the form of silver oxide, not metallic silver it seems. Why oxygen was used, you might use Ar?. In order to confirm it, FTIR or XPS studies can be performed if possible.

5.Before the actual deposition of Ag, any surface cleaning/etching process was done?.

6.Line 131: L= 80.2 mm?

7.Line 208: N1 e N2?

8.Line 240: EC and EAg resulted ?? times higher

9.Fig. 5: Fig. 5 a is uncoated sample. Thus…..’Blue lines  indicate fibre-like regions which resulted conformally coated by the Ag nanostructure’, needs a modification.

10.Line 308: so that h represents…/?

  1. Fig. 7: What is the initial and final weight of the sample for TGA?, It seems after 400°C, formation of char begins.

Author Response

The manuscript ‘Impact of surface functionalization by nanostructured silver thin films on thermoplastic central venous catheters: mechanical, microscopical and thermal analyses’ is interesting and written well. It can be accepted after a minor correction based on the below comments.

We thank the Reviewer for the positive evaluation. In the following, we provide a point-by-point reply to all the raised concerns.

1.Line 51: of obtaining…

2.Line 64: Ag0…0 can be in superscript, or else it looks like silver oxide.

These items have now been corrected

3.Sec. 2.2.: The coating is done on the top only or including in the septum region also?

The coating was deposited on the external surface of the CVC, except the septum; this circumstance has now been better highlighted in the main text (Line 149 and Caption of Fig. 2)

4.Line 109: The oxygen was used in the chamber thus the coating may be in the form of silver oxide, not metallic silver it seems. Why oxygen was used, you might use Ar?. In order to confirm it, FTIR or XPS studies can be performed if possible.

We thank the Reviewer for this remark; the reasons for using oxygen instead of Ar during deposition have been detailed in our previous work on PEA-deposition of silver (Ref. 30 in the main text); in brief, using Oxygen improves importantly the quality of the films and their morphological characteristics, especially in view of obtaining, as desired, nanostructuration of the surface. Being aware of the importance to distinguish between either metallic and oxidized silver, we also carried out an extensive XPS characterization with determination of the Auger parameter: within the resolution limit of XPS, we found that only metallic Ag0 was obtained. We have now better explained this circumstance (Lines 133-137).

5.Before the actual deposition of Ag, any surface cleaning/etching process was done?

We thank the reviewer for the valuable comment. The authors want to clarify that CVCs were received in sealed package from the manufacturer – i.e. B. Braun Avitum Italy S.r.l – after sterilization in accordance to the relative normative of such medical device. Thus, no further preparation of the surface was required before inserting the specimens in the vacuum chamber for the deposition. On the other hand, any surface treatment which could have been detrimental for the integrity of the surface was avoided. Accordingly, we have now better specified this circumstance in the manuscript (Lines 138-142).

6.Line 131: L= 80.2 mm?

7.Line 208: N1 e N2?

8.Line 240: EC and EAg resulted?? times higher

We apologize for these typos, which have now been corrected.

9.Fig. 5: Fig. 5 a is uncoated sample. Thus…..’Blue lines indicate fibre-like regions which resulted conformally coated by the Ag nanostructure’, needs a modification.

The above sentence has been modified to improve its clarity (see Caption of Fig. 5).

10.Line 308: so that h represents…/?

This part has been entirely rewritten to improve clarity (Lines 375-380)

11. Fig. 7: What is the initial and final weight of the sample for TGA?, It seems after 400°C, formation of char begins.

In the main text it is specified that 15 mg were used (Line 239). Accordingly, 15 mg is the initial weight. In the curve TG the weight loss% is represented (Fig. 7), so that the final weight can be easily calculated.

Round 2

Reviewer 2 Report

The responses of the authors are enough good, the improvement of the paper is adequate after revision and the paper could be published without further modifications.